# Convalescent Plasma Efficacy in Life-Threatening COVID-19 Patients Admitted to the ICU: A Retrospective Cohort Study

**DOI:** 10.3390/jcm10102113

**Published:** 2021-05-14

**Authors:** Mohamed Abuzakouk, Khaled Saleh, Manuel Algora, Ahmad Nusair, Jawahir Alameri, Fatema Alshehhi, Sara Alkhaja, Mohamed Badr, Khaled Abdallah, Bruno De Oliveira, Ashraf Nadeem, Yeldho Varghese, Dnyaseshwar Munde, Shameen Salam, Baraa Abduljawad, Hussam Elkambergy, Ali Wahla, Ahmed Taha, Jamil Dibu, Ahmed Bayrlee, Fadi Hamed, Laila AbdelWareth, Nadeem Rahman, Jorge Guzman, Jihad Mallat

**Affiliations:** 1Division of Rheumatology, Department of Internal Medicine, Cleveland Clinic Abu Dhabi, Abu Dhabi 112412, United Arab Emirates; 2Cleveland Clinic Lerner College of Medicine, Case Western Reserve University, Cleveland, OH 44195, USA; AlgoraM@ClevelandClinicAbuDhabi.ae (M.A.); wahlaa@clevelandclinicabudhabi.ae (A.W.); 3Critical Care Institute, Cleveland Clinic Abu Dhabi, Abu Dhabi 112412, United Arab Emirates; salehk@clevelandclinicabudhabi.ae (K.S.); rashadicu@gmail.com (M.B.); Dr_khaled_Salah@windoslive.com (K.A.); deolivb@clevelandclinicabudhabi.ae (B.D.O.); ashmohnad@hotmail.com (A.N.); VargheY2@ClevelandClinicAbuDhabi.ae (Y.V.); MundeD@ClevelandClinicAbuDhabi.ae (D.M.); SalamS3@ClevelandClinicAbuDhabi.ae (S.S.); AbduljB@ClevelandClinicAbuDhabi.ae (B.A.); ElkambH@ClevelandClinicAbuDhabi.ae (H.E.); TahaA2@ClevelandClinicAbuDhabi.ae (A.T.); DibuJ@ClevelandClinicAbuDhabi.ae (J.D.); BayrleA@ClevelandClinicAbuDhabi.ae (A.B.); HamedF@ClevelandClinicAbuDhabi.ae (F.H.); RahmanN2@ClevelandClinicAbuDhabi.ae (N.R.); GuzmanJ@ClevelandClinicAbuDhabi.ae (J.G.); 4Pathology and Laboratory Medicine Institute, Cleveland Clinic Abu Dhabi, Abu Dhabi 112412, United Arab Emirates; WarethL@ClevelandClinicAbuDhabi.ae; 5Medical Subspecialties Institute-Infectious Diseases, Cleveland Clinic Abu Dhabi, Abu Dhabi 112412, United Arab Emirates; NusairA@clevelandclinicabudhabi.ae; 6Education Institute, Cleveland Clinic Abu Dhabi, Abu Dhabi 112412, United Arab Emirates; Alamerj3@clevelandclinicabudhabi.ae (J.A.); AlshehF2@clevelandclinicabudhabi.ae (F.A.); AlkhajS3@clevelandclinicabudhabi.ae (S.A.); 7Faculty of Medicine, Normandy University, UNICAEN, ED 497, 1400 Caen, France

**Keywords:** convalescent plasma, life-threatening COVID-19, critically ill, mechanical ventilation, intensive care unit, neutralizing antibody, SARS-CoV-2

## Abstract

(1) Background: There are limited data regarding the efficacy of convalescent plasma (CP) in critically ill patients admitted to the intensive care unit (ICU) due to coronavirus disease 2019 (COVID-19). We aimed to determine whether CP is associated with better clinical outcome among these patients. (2) Methods: A retrospective single-center study including adult patients with laboratory-confirmed SARS-CoV-2 infection admitted to the ICU for acute respiratory failure. The primary outcome was time to clinical improvement, within 28 days, defined as patient discharged alive or reduction of 2 points on a 6-point disease severity scale. (3) Results: Overall, 110 COVID-19 patients were admitted. Thirty-two patients (29%) received CP; among them, 62.5% received at least one CP with high neutralizing antibody titers (≥1:160). Clinical improvement occurred within 28 days in 14 patients (43.7%) of the CP group vs. 48 patients (61.5%) in the non-CP group (hazard ratio (HR): 0.75 (95% CI: 0.41–1.37), *p* = 0.35). After adjusting for potential confounding factors, CP was not independently associated with time to clinical improvement (HR: 0.53 (95% CI: 0.23–1.22), *p* = 0.14). Additionally, the average treatment effects of CP, calculated using the inverse probability weights (IPW), was not associated with the primary outcome (−0.14 days (95% CI: −3.19–2.91 days), *p* = 0.93). Hospital mortality did not differ between CP and non-CP groups (31.2% vs. 19.2%, *p* = 0.17, respectively). Comparing CP with high neutralizing antibody titers to the other group yielded the same findings. (4) Conclusions: In this study of life-threatening COVID-19 patients, CP was not associated with time to clinical improvement within 28 days, or hospital mortality.

## 1. Background

Since December 2019, the COVID-19 pandemic has claimed more than 2.5 million lives around the world [1]. The disease continues to spread rapidly across the globe, with the CDC reporting at the time of writing this paper about 75,000 new cases per day in the US alone [2]. Despite some advances in drug therapy, namely, remdisivir in patients with mild disease and dexamethasone in hospitalized patients on oxygen, the treatment remains far from ideal [3,4].

Convalescent Plasma (CP) from recovered patients, believed to provide passive immunity against viral infections, was tried more than a hundred years ago during the Spanish flu [5]. In the last fifteen years, it has resurfaced again as a potential treatment of many viral illnesses, including SARS, MERS with inconclusive results [6,7]. Over the previous several months, there has been mounting evidence for CP’s use in patients with COVID-19. The creation of the Expanded Access Program funded by the US government resulted in an open-label, observational study of 35,000 COVID-19 patients treated with CP [8]. A retrospective study based on this registry that included 3082 patients showed a significant reduction in mortality among patients who received the CP treatment with higher antibody titers compared to CP with lower antibody levels [9]. This was also confirmed by a randomized trial, which found that early administration of higher-titer CP to elderly outpatients with mild disease reduced the progression of COVID-19 to severe illness [10]. However, a randomized study in India of 464 hospitalized patients with moderate disease, using CP administration but at low titers, yielded a negative result [11]. Two other randomized trials of patients with severe disease, did not show survival benefits or reduction in progression to severe COVID-19 disease [12,13]. The latter reached only 50% of its planned enrollment due to the decrease in the number of patients admitted with COVID-19, which could have affected the outcome [13]. A recent meta-analysis that included 10 RCTs showed no association between CP and all-cause mortality or other clinical outcomes in COVID-19 patients [14]. Although these trials did not show a consistent improvement in outcome, which is most likely related to the patient population, severity of illness, time of administration, and the plasma antibody titer, they all did show that CP is safe and well-tolerated. These trials included few or no patients with life-threatening COVID-19 disease requiring high flow oxygen therapy (HFNO), non-invasive ventilation (NIV), or invasive mechanical ventilation (IMV). Data focusing on the critically ill COVID-19 patients in the intensive care unit (ICU) are still scarce [15,16]. Thus, the benefits of CP in these patients are still not known. Our study aimed to evaluate the efficacy of CP in patients with life-threatening COVID-19 disease admitted to the ICU.

## 2. Methods

This retrospective study was approved by the Institutional Ethics Committee of Cleveland Clinic Abu Dhabi (REC number: A-2020-029) and waived the need for informed consent due to the retrospective nature of the study.

All adult patients (age ≥ 18 years) were admitted to our ICU between 1 March 2020 and 29 May 2020, with confirmed SARS-CoV-2 infection (virus detected by a real-time reverse-transcriptase–polymerase-chain-reaction assay of a nasopharyngeal sample) and life-threatening COVID-19 pneumonia confirmed by chest imaging were included. Life-threatening COVID-19 was defined as acute respiratory failure requiring mechanical ventilation (invasive or non-invasive) or high flow oxygen therapy, shock, or other organ failure requiring ICU monitoring. Patients were excluded if they had a brief (less than 24 h) ICU stay or if the COVID-19 infection was deemed to be incidental and did not impact their ICU admission.

### 2.1. Outcome Measures

The primary end point was time to clinical improvement within a 28-day period. Clinical improvement was defined as patient discharge or a reduction of 2 points on a 6-point disease severity scale [17]. The scale was defined as follow: 6 points, death; 5 points, hospitalization plus extracorporeal membrane oxygenation (ECMO) or invasive mechanical ventilation (IMV); 4 points, hospitalization plus NIV or HFNO; 3 points, hospitalization plus supplemental oxygen (not HFNO or NIV); 2 points, hospitalization with no supplemental oxygen; 1 point, hospital discharge.

Secondary clinical outcomes were as follows: hospital mortality rate, ICU and hospital length of stays, duration of IMV, rates of improvement at days 14 and 28, and conversion of a nasopharyngeal swab of viral PCR results from positive at baseline to negative at day 14.

Data were collected on baseline characteristics including demographics, and the presence of medical comorbidities. Laboratory values, including full blood count, coagulation parameters, and inflammatory markers (C-reactive protein, interleukin 6, and ferritin), were collected on admission to ICU. The need and duration of IMV, the use of vasopressors, renal replacement therapy, and ECMO, anti-viral drugs, and steroids were also collected.

### 2.2. Convalescent Plasma Procurement

The selection criteria for donors included all the requirements for the prevention of transfusion-transmitted diseases and, in addition, required a minimum period of 14 days after resolution of the respiratory disease, as evidence of a complete recovery before donation. Furthermore, to guarantee donors’ safety, no more than two collections of a maximum of 600 mL (400 mL if <80 kg of weight) were allowed to be carried out separated by an interval of at least two weeks. Plasma collection and preparation were performed at the Abu Dhabi Blood Bank (ADBB), a certified collection facility that is routinely dedicated to donation following international guidance and accredited by the American Bank Association of Blood. Apheresis procedure (plasmapheresis) is nowadays the preferred methodology for plasma collection. To minimize the risk of transfusion-related acute lung injury, only male or non-pregnant women were selected as the plasma source. Thus, the products obtained at ADBB were frozen during the first 8 h of collection, following good manufacturing guidelines. Fresh frozen plasma was transported to Cleveland Clinic Abu Dhabi, maintaining its solid state, stored at −20 °C, and thawed at 37 °C for immediate transfusion.

A total of 36 valid donors were identified. Forty-five apheresis procedures were performed, with no remarkable secondary effects. In the end, a total of 90 CP products were collected.

The decision of CP administration was based on our hospital protocol for COVID-19 management. Patients with severe COVID-19 infection requiring low oxygen therapy, HFNO, NIV, or IMV were candidates for CP treatment. Exclusion criteria were patients under the age of 18, pregnant women, and known IgA deficiency.

### 2.3. Antibody Titer Determination

Antibodies were determined by LIAISON^®^ SARS-CoV-2 S1/S2 IgG (DiaSorin- Italy), which uses chemiluminescence immunoassay (CLIA) technology for the quantitative determination of anti-S1 and anti-S2 specific IgG antibodies to SARS-CoV-2 in human serum or plasma samples. A cut-off of >15 AU/mL is resulted as positive, whereas <12 AU/mL is considered negative. Values between 12–15 AU/mL are considered equivocal or in the seroconversion phase. The assay is intended to support the study of the infected patient’s immune status by indicating the presence of neutralizing IgG antibodies against SARS-CoV-2. Positive results have a concordance of 94.4% with neutralizing antibodies. A cut-off value > 80 AU/mL on this assay correlate with neutralizing antibody titer levels of 1:160 on the neutralization test assay [18].

Antibody titer against SARS-CoV-2 assessment in the CP was done after CP was administered to the patients. Seven donors were found to have CP with negative antibody titers, and the patients who received those products were excluded from the analysis. In the rest (29 donors), IgG antibodies against SARS-CoV-2 were positive (>15 AU/mL). In 62.5% of the cases, the titer was >80 AU/mL, corresponding with neutralizing antibody titer levels of ≥1:160.

### 2.4. Statistical Analysis

The normality of data distribution was assessed using the Shapiro–Wilk test and by visually checking each variable’s distribution (histogram). Data are expressed as mean ± SD when normally distributed or as median [IQR] when non-normally distributed. Proportions were used as descriptive statistics for categorical variables. Comparisons of values between independent groups were performed by the 2-tailed Student *t* test or the Mann–Whitney *U* test, as appropriate. Analysis of the discrete data was performed by *χ*2 test or Fisher exact test when the numbers were small. There were missing data (missing at random) for interleukin 6 (3.6%), ferritin (1.8%), D-dimer (1%), fibrinogen (8.2%), APTT (9.1%), and INR (9.1%) that were imputed using multiple imputations using 50 imputed datasets.

Time-to-event data were analyzed using the Kaplan–Meier method, and a Log-rank test was used to compare outcomes of patients who received convalescent plasma and those who did not receive the drug. For the primary endpoint of time to clinical improvement, death before day 28 was considered to be right censored at the last observation date. Adjusted Cox proportional hazards regression models were fitted to estimate the association between convalescent plasma and time to clinical improvement, using clinically likely confounders including age, gender, body mass index, comorbidities, SOFA/SAPS II scores, IMV, renal replacement therapy (RRT), ECMO, use of vasopressors, steroids, and tocilizumab. In addition, variables associated with time to clinical improvement (*p* < 0.1) in univariate analysis were also included in the adjusted Cox model. The potential problem of co-linearity was evaluated using Spearman or Pearson correlation coefficient before running the analysis. Proportionality hazard assumption was assessed using the Schoenfeld residuals. Hazard ratios (HRs) and 95% confidence intervals were summarized.

To estimate the average treatment effects (ATE) of the CP, we used the inverse probability weights (IPW) with robust standard error using the survival treatment effects function in Stata (steffects ipw). The same confounding variables (as in the adjusted Cox model) were used to estimate the parameters of the treatment-assignment model and the time-to-censoring model. The balance of confounding variables between the CP and non-CP groups were assessed by calculating the weighted standardized differences. There is no rule regarding how much imbalance is acceptable in a propensity score. Proposed maximum standardized differences for specific covariates range from 10 to 25% [19,20,21]. Statistical analyses were performed using Stata/SE software version 16.0 for Windows (Stata Corp LLC, College Station, TX, USA). A value of *p* < 0.05 was considered statistically significant, and all reported *p*-values are two-sided.

## 3. Results

### 3.1. Study Population

From March 1st to 29 May 2020, one-hundred and ten adult patients with acute respiratory failure caused by COVID-19 infection were admitted to the ICU and were included in this study (Figure 1). The main characteristics of the cohort are summarized in Table 1. The median age among all patients was 49 years (IQR: 40–58 years), and 99 (90%) were men. Among the patients, 70 (63.6%) had at least one comorbidity, 76 (69%) received IMV, 68 (62%) required vasopressor support, and 28 (25.5%) received RRT. All our included patients were immunocompetent, none had any cancer or was on immunosuppressive treatment. The median time from symptoms onset to ICU admission was 5 days (IQR: 3–7 days).

Thirty-two patients (29%) received CP; among them, nine (28%) received a second dose of plasma infusion. The median plasma infusion volume was 200 mL (IQR: 200–400 mL). Twenty (62.5%) out of the 32 patients received at least one CP with significant neutralizing antibody titers (≥1:160). The median time from symptoms onset to CP administration was 6 days (IQR: 4–11 days), and the median time from ICU admission to CP infusion was 3 days (IQR: 2–7 days). Patients’ characteristics, comorbidity, and severity scores were not significantly different between the CP and non-CP groups (Table 1). The time from symptoms onset to ICU admission was significantly shorter in the CP group than in the non-CP group. Regarding vital signs on ICU admission, only respiratory rate was significantly higher in patients who received CP than patients who did not receive CP. There were no significant differences between the two groups regarding laboratory data except for INR, which was significantly higher in the non-CP group (Table 1). The treatments received in ICU (IMV, vasopressor support, RRT, ECMO, antiviral drugs, Tocilizumab, and steroids) did not differ significantly between the two groups except for the use of NIV, which was significantly higher in the CP group (Table 1).

### 3.2. Primary Clinical Outcome

Overall, 62 patients (56.5%) experienced clinical improvement by day-28. In univariate analysis (without adjustment), CP was not associated with time to clinical improvement (HR = 0.75 (95%CI: 0.41–1.37), *p* = 0.35) (Figure 2 and Appendix A).

In the multivariable Cox regression analysis, after adjusting for the clinically known confounding variables (mentioned in the methods) along with time from symptoms onset to CP administration, leucocytes count, D-dimer, lactate, aspartate aminotransferase levels, and PaO2/FiO2 ratio, exposure to CP was still not significantly associated with time to clinical improvement (HR = 0.53 (95%CI: 0.23–1.22), *p* = 0.14) (Table 2). The global test of proportional-hazards assumption for the model was not statistically significant (*p* = 0.20), testifying the goodness of fit of the model. Replacing SAPS II score by SOFA score (we did not include SAPS II and SOFA scores in the same model because of the strong correlation between the two variables, r = 0.77, *p* < 0.001) in multivariable model yielded the same findings. Additionally, CP with significant neutralizing antibodies titers (≥1:160) was not associated with time to clinical improvement (HR = 1.05 (95%CI: 0.40–2.72), *p* = 0.92) (Appendix A).

We calculated the ATE of CP based on the ipw estimators. We found that the ATE of CP was also not significantly related to the time to clinical improvement by ((−0.14 days (95% CI: −3.19–2.91 days), *p* = 0.93). The same covariates that were included in the multivariable Cox regression analysis model were used for ipw estimators. The weighted standardized differences were <20% for all the covariates (Appendix A). Additionally, the ATE of CP with high neutralizing antibody titers (≥1:160) was not significantly associated with the time to clinical improvement by ((−1.55 days (95% CI: −4.53–1.44 days), *p* = 0.31).

### 3.3. Secondary Clinical Outcomes

There were no significant differences between CP and non-CP groups regarding hospital mortality rate (31.2% vs. 19.2%, *p* = 0.17, respectively), median of ICU length of stay (15 days (IQR: 9–28 days) vs. 14 days (IQR: 6–26 days), *p* = 0.45, respectively), and median of hospital length of stay (19 days (IQR: 14–34 days) vs. 25 days (IQR: 15–43 days), *p* = 0.31, respectively). Additionally, the duration of IMV did not differ between the two groups (9 days (IQR: 0–22 days) for the CP group vs. 9 days (IQR: 0–23 days) for the non-CP group, *p* = 0.75). On day 28, only 14 patients (43.7%) experienced a clinical improvement in the CP group compared to 48 (61.5%) in the non-CP group but did not reach statistical significance (*p* = 0.09). Additionally, 9 patients (28.1%) experienced clinical improvement on day 14 in the CP group compared to 21 (26.9%) in the non-CP group (*p* = 0.89).

On day 14, only 13 patients (40.6%) who received CP had their SARSCoV-2 tests turned negative compared to 42 patients (53.8%) who did not receive CP (*p* = 0.21). Additionally, among the 20 patients who received at least 1 convalescent plasma with higher neutralizing antibody titer levels (≥1/160), 11 (55%) had their SARSCoV-2 tests turned negative on day 14 compared to 44 patients (48.9%) in the other group (*p* = 0.62).

CP use was well tolerated in our patients; we did not observe any side effects.

## 4. Discussion

In this study of patients with life-threatening COVID-19 disease, CP treatment was not associated with the time to clinical improvement using the six-points ordinal scale ranging from recovery to death. There were no differences between the CP treatment group and the standard therapy group regarding the hospital mortality rate, ICU, and hospital length of stays or duration of invasive mechanical. Additionally, CP with higher antibody titer levels did not show an association with the primary outcome. Furthermore, CP therapy was not related to higher rates of negative SARS-CoV-2 viral PCR results from nasopharyngeal swabs on day 14, suggesting that CP treatment was not associated with antiviral activity in life-threatening COVID-19.

In a randomized controlled trial of 228 patients with severe COVID-19 pneumonia, CP administration was not associated with time to improvement by using a similar six-point ordinal scale, even though more than 95% of the transfused CP units had a total antibody titer of at least 1:800 (COVIDAR method) [12]. However, most of their patients (65%) were on the simple nasal cannula, and most of the patients received CP later after the symptoms’ onset (median of 8 days). Nevertheless, no differences in favor of CP were noted in the subgroup of patients who received the intervention within 72 h after the onset of symptoms [12]. Another randomized controlled trial in patients with moderate COVID-19 showed no difference in severe disease or death, although the infused CP had very-low titers of specific antibodies [11].

A very few studies focused on critically ill COVID-19 patients in ICU requiring HFNO and/or mechanical ventilation [15,16]. Kurtz et al., in a retrospective study of 113 life-threatening COVID-19 patients with acute respiratory failure (86% received mechanical ventilation) in ICU, found no association between CP administration and clinical improvement or death compared to standard of care management [15]. Additionally, no difference in respiratory improvement between CP treatment and standard of care was found in another retrospective study that included 80 critically ill COVID-19 patients, of whom 86% required IMV [16]. However, no information about the antibody titer levels in the infused CP was provided in the two studies; additionally, the duration from symptoms’ onset to CP administration was long (median 13 days and 10 days, respectively) [15,16]. In a randomized, open-label clinical trial of CP treatment in 103 patients with severe or life-threatening COVID-19 pneumonia, Li et al. found no differences in the time to clinical improvement or 28-day mortality in comparison with placebo [13]. However, patients received CP even much later after symptoms onset (median 14 days). Additionally, the study was stopped earlier due to a drop in enrollment which could have affected the results. Our results align with these findings, even though the time from the onset of symptoms to CP administration was shorter in our study (median 6 days). Moreover, CP treatment with higher neutralizing antibody titers was not related to a better outcome. In a retrospective study that included 3082 COVID-19 hospitalized patients who received CP, the authors showed that patients CP with higher antibody titers (>18.45, VITROS method) had lower 30 days mortality than those who received lower titers, but not in patients who were receiving mechanical ventilation [9].

There are several possible explanations for the lack of clinical benefit of CP in our study. First, our patients were in the advanced stage of the disease, which is mainly driven by markedly elevated proinflammatory cytokines and chemokines [22,23,24]. Thus, neutralizing antibodies from CP may not be adequate to reverse the hyperinflammatory state and leads to clinical improvement [25]. Indeed, earlier administration of CP within 72 h after the onset of mild COVID-19 symptoms in older patients resulted in reduced progression to severe COVID-19 disease [10]. Second, in an open-label randomized, controlled clinical trial that was stopped early, 79% of the patients tested had detectable neutralizing antibodies, with median titers similar to those of the donors, on the day of inclusion (median 10 days from symptoms’ onset) [26]. This is an important factor that CP is most likely to be helpful when given close to the disease onset where the patient has not developed sufficient innate neutralizing antibody.

In contrast to previous reports [13], in our study, CP, even with significant neutralizing antibody titers, was not associated with a higher viral clearance compared to standard of care management. Our results are in line with a recent study that found no difference in viral clearance rate between CP and standard of care in critically ill patients [16]. Furthermore, Kurtz et al. found that only 46% of life-threatening COVID-19 patients had negative SARS-CoV-2 PCR tests on day 14 after CP administration, which is in agreement with our findings (40.6% on day 14) [15]. Therefore, the incapability of CP treatment to increase the viral clearance might be due to the severity of COVID-19 patients, which may have prevented any potential benefit of CP.

We did not observe any mild or serious transfusion-related adverse events in our patients. This is in agreement with a large cohort of COVID-19 patients receiving CP, where serious complications were rare (<1%) [14,27].

A large proportion of our patients (85.5%) received tocilizumab (anti-interleukin-6 receptor) according to our hospital protocol treatment for COVID-19 patients (pneumonia with bilateral involvement associated with elevated interleukin-6 and C-reactive protein, requiring oxygen therapy). Recently, in a randomized, controlled, and open-label trial (RECOVERY), tocilizumab was associated with improved survival and other clinical outcomes in severe COVID-19 patients [28]. However, we do not think that the high prescription rate of tocilizumab would have influenced our findings. Indeed, tocilizumab was equally distributed between the CP and non-CP groups (Table 1), and it was adjusted for in the multivariable Cox regression analysis (Table 2) and used to determine ipw estimators (Appendix A).

Our results are of clinical importance and add significant data to CP’s existing literature in life-threatening COVID-19 patients. CP was, on average, given relatively early after symptoms onset in a rapidly deteriorating disease process, further distinguishing this group from the other studies [13,15,16]. We used robust statistical tools to account for confounding variables and minimize other treatment effects. At the time of writing this article, there is ongoing multicenter RCT to assess the efficacy of CP therapy in critically ill COVID-19 patients treated with invasive mechanical ventilation for acute respiratory failure (the CONFIDENT trial) [29]. The results of this trial will shed further light on the effects of CP on the disease process and the clinical outcomes of COVID-19 life-threatening patients.

With the advent of highly effective vaccination against COVID-19, it is unlikely that CP will have a therapeutic role in immune-competent vaccinated patients who fall to the disease. It is feasible that CP might still play a role in non-vaccinated patients or immunocompromised patients who might not mount the necessary immune response to the vaccine. In this regard, it has been suggested that CP might provide at least a modest benefit in seronegative COVID-19 patients [30]. This might be particularly the case in vulnerable patient populations (elderly and/or patients with significant comorbid conditions) who are rapidly deteriorating within few days of symptoms onset [10].

Our study has some important limitations. First, it is a single-center retrospective study conducted at a quaternary care facility in the Middle East. Thus, management and outcomes do not necessarily reflect those at other centers. Second, CP antibody titer assessment was performed after the plasma was administered to the patients. Additionally, neutralizing antibody titer activity against SARS-CoV-2 was not directly measured but determined based on the relationship between the IgG titer levels and the neutralizing antibody titer levels (concordance 94.4%) [18]. Third, the relatively low number of patients with high neutralizing antibody titer could have affected the power to detect a difference in clinical outcome. Fourth, there was no measurement of antibody titer levels of neutralizing antibody titers against SARS-CoV-2 in the recipient population before and after receiving CP. Additionally, many patients that were candidates for CP therapy did not receive the treatment because of lack of availability (based on ABO blood type matching between donors and patients). However, this was not uncommon at the time of the treatment during the early phase of the pandemic. Fifth, despite multivariable analysis and adjustment for potential confounders, we cannot rule out bias selection or residual confounding.

## 5. Conclusions

In this retrospective study of life-threatening COVID-19 patients admitted to ICU, CP was not associated with the time to clinical improvement within 28 days, hospital mortality, or increased viral clearance.

## Figures and Tables

**Figure 1 jcm-10-02113-f001:**
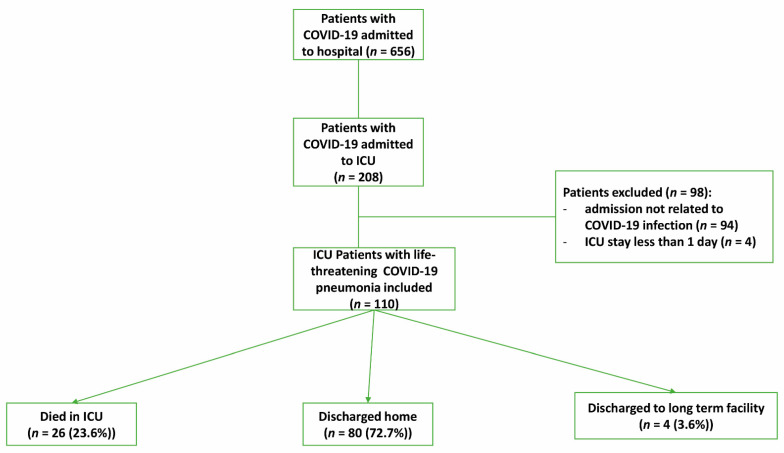
Flow chart of COVID-19 patients admitted to the intensive care unit and their outcomes. ICU, intensive care unit.

**Figure 2 jcm-10-02113-f002:**
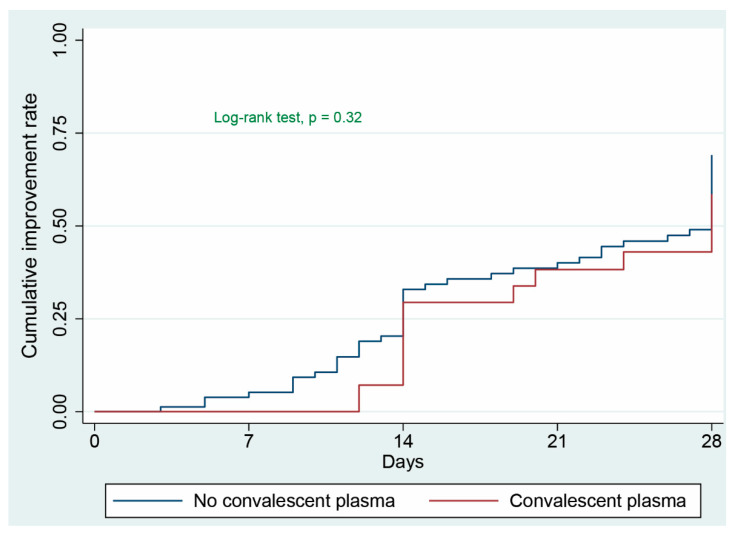
The clinical improvement rate is the ratio of patients who experienced a 2-point improvement or were discharged alive from the hospital. All patients who did not reach clinical improvement were observed for the full 28-day period or until death. The median time to clinical improvement was 28 days (95% CI: 14–indeterminate days) in patients who received convalescent plasma therapy compared to 28 days (95% CI: 19–28 days) in patients who received only standard care treatment (HR = 0.75 (95%CI: 0.41–1.37), *p* = 0.35).

**Table 1 jcm-10-02113-t001:** Comparisons of baseline characteristics, laboratory data, treatments, and outcomes between convalescent plasma (CP) and non-CP groups.

Variables	All Patients (*n* = 110)	Convalescent Plasma (*n* = 32)	Non-Convalescent Plasma (*n* = 78)	*p*-Value
Age, year	49 (40–58)	50 (43–60)	46 (39–57)	0.22
Male, *n* (%)	99 (90)	29 (90.6)	70 (89.7)	1.00
Body mass index, kg·m^−2^	26.2 (23.5–30.6)	25.7 (22.4–27.5)	27.0 (23.7–31.1)	0.13
SOFA score	5 (3–8)	5 (3–7.5)	5.5 (3–9)	0.44
SAPS II score	31 (24–44)	30 (23–44)	31 (24–45)	0.95
Patients with comorbidities, *n* (%)	70 (63.6)	23 (71.9)	47 (60.3)	0.28
Comorbidities distribution, *n* (%)				
Diabetes mellitus	48 (43.6)	13 (40.6)	35 (44.9)	0.68
Hypertension	45 (40.9)	14 (43.7)	31 (39.7)	0.70
Chronic artery disease	8 (7.3)	2 (6.2)	6 (7.7)	1.00
Chronic kidney disease	7 (6.4)	1 (3.1)	6 (7.7)	0.67
Time from symptoms to ICU admission, day	5 (3.2–7)	4 (3–6)	5 (4–8)	0.04
Vital signs on hospital/ICU admission, day				
Temperature (max) ≥ 38 °C, *n* (%)	48 (43.6)	16 (50.0)	32 (41.0)	0.39
Heart rate (max), beats·min^−1^	105 ± 19	107 ± 20	104 ± 19	0.44
Respiratory rate (max), breaths·min^−1^	32 ± 8	36 ± 8	31 ± 8	0.004
Laboratory data on ICU admission				
C-reactive protein, mg·L^−1^	139 (63–225)	121 (49–210)	141 (64–238)	0.25
Leucocytes count, ×10^9^·L^−1^	8.9 (6.3–11.9)	8.4 (5.3–11.3)	9.3 (7.1–12.0)	0.22
Lymphocytes count, ×10^9^·L^−1^	0.77 (0.48–1.09)	0.75 (0.41–1.04)	0.77 (0.49–1.12)	0.57
Lymphocytes ≤ 1 × 10^9^·L^−1^; *n* (%)	80 (72.7)	24 (75.0)	56 (71.8)	0.82
Platelet count, ×10^9^·L^−1^	224 (164–298)	216 (152–298)	228 (180–299)	0.51
Procalcitonin, ng·mL^−1^	0.49 (0.20–3.05)	0.40 (0.16–1.05)	0.58 (0.21–3.71)	0.16
INR	1.2 (1.1–1.2)	1.1 (1.1–1.2)	1.2 (1.1–1.3)	0.044
Activated partial thromboplastin time; s	34.5 (30.2–35.7)	34.3 (31.6–38.3)	35.2 (30.0–39.9)	0.97
D-dimer, µg·mL^−1^ (normal reference: < 0.05)	2.6 (0.9–4.0)	1.8 (0.7–4.0)	3.0 (1.2–4.0)	0.10
D-dimer > 2 µg·mL^−1^, *n* (%)	64 (58.2%)	15 (46.9)	49 (62.8)	0.14
Fibrinogen, g·L^−1^	6.0 (4.8–7.0)	5.8 (4.7–6.4)	6.3 (5.0–7.2)	0.31
Ferritin, µg·L^−1^ (reference range: 36–480)	1519 (793–2481)	1538 (923–2388)	1484 (749–2501)	0.93
Interleukin 6, ng·L^−1^	219 (103–899)	181 (123–823)	234 (96–904)	0.85
Alanine aminotransferase, IU·mL^−1^	38 (26–64)	40 (26–73)	37 (25–60)	0.34
Aspartate aminotransferase, IU·mL^−1^	53 (33–91)	52 (35–84)	53 (32–91)	0.89
Total bilirubin, µmol·L^−1^	10.7 (7.5–16.1)	9.3 (6.9–15.7)	11.1 (8.1–16.7)	0.15
Creatinine, µmol·L^−1^	78 (64–144)	73 (62–89)	82 (66–167)	0.12
PaO_2_/FiO_2_ ratio, mmHg	85 (66–144)	85 (67–121)	95 (64–158)	0.41
Lactate levels, mmol·L^−1^	1.4 (1.2–1.7)	1.5 (1.2–1.8)	1.4 (1.2–1.7)	0.35
Treatments, *n* (%)				
Invasive mechanical ventilation	76 (69.1)	21 (65.3)	55 (70.5)	0.61
High flow nasal oxygen therapy	53 (48.2)	20 (62.5)	33 (42.3)	0.05
Non-invasive ventilation	44 (40)	19 (59.4)	25 (32.0)	0.008
Vasopressor support	68 (61.8)	21 (65.6)	47 (60.3)	0.60
Renal replacement therapy	28 (25.5)	6 (18.7)	22 (28.2)	0.30
Extracorporeal membrane oxygenation	9 (8.2)	3 (9.4)	6 (7.7)	0.72
Tocilizumab	94 (85.5)	27 (84.4)	67 (85.9)	1.00
Hydroxychloroquine	45 (40.9)	10 (31.2)	35 (44.9)	0.19
Favipiravir	28 (25.5)	8 (25.0)	20 (25.6)	1.00
Lopinavir/ritonavir	30 (27.3)	7 (21.9)	23 (29.5)	0.49
Methylprednisolone	42 (38.2)	15 (46.9)	27 (34.6)	0.23
WHO 6-point disease severity scale on ICU admission, *n* (%)				
Scale 4	53 (48.2)	21 (65.6)	36 (46.1)	0.06
Scale 5	57 (51.8)	11 (34.4)	42 (53.8)	

SOFA, Sequential Organ Failure Assessment; SAPS, Simplified Acute Physiology Score; ICU, intensive care unit; PaO_2_, arterial oxygen pressure; FiO_2_, inspiratory oxygen fraction; INR, international normalized ratio; WHO, World Health Organization. Data are reported as median (IQR), mean ± SD, or number (proportion).

**Table 2 jcm-10-02113-t002:** Multivariable Cox regression analysis to predict time to clinical improvement.

Variables	Hazard Ratio	95% Confidence Interval	*p*-Value
Convalescent plasma therapy (refer: no)	0.53	0.23–1.22	0.14
Age, year	0.98	0.95–1.00	0.15
Male, (refer: female)	1.16	0.28–4.74	0.83
BMI, kg·m^−2^	0.97	0.90–1.04	0.38
SOFA score	1.03	0.90–1.18	0.68
Lactate, mmol·L^−1^	0.76	0.40–1.44	0.40
Leucocytes count, ×10^9^·L^−1^	1.00	0.93–1.09	0.88
PaO_2_/FiO_2_ ratio, mmHg	1.00	0.99–1.00	0.79
D-dimer, µg·mL^−1^	0.95	0.69–1.31	0.77
Aspartate aminotransferase, IU·mL^−1^	0.99	0.98–0.99	0.03
Time from symptoms onset to convalescent plasma infusion, day	1.00	0.92–1.10	0.85
Invasive mechanical ventilation, (refer: HFNO/NIV)	0.59	0.23–1.50	0.27
Vasopressor support, (refer: no)	0.43	0.20–0.96	0.04
Renal replacement therapy, (refer: no)	0.36	0.12–1.08	0.07
Methylprednisolone, (refer: no)	1.05	0.58–1.91	0.87
Tocilizumab, (refer: no)	1.02	0.34–3.10	0.97
Extracorporeal oxygenation membrane, (refer: no)	-	-	-
Comorbidities, (refer: no)	0.82	0.45–1.49	0.51

HFNO, high flow nasal oxygen; NIV, non-invasive ventilation; PaO_2_, arterial oxygen pressure; FiO_2_, inspiratory oxygen fraction; SOFA, Sequential Organ Failure Assessment. Test of proportional-hazards assumption (Shoenfeld residuals test): *p* = 0.20.

## Data Availability

The data presented in this study are available on request from the corresponding author. The data are not publicly available due to the Ethics Committee restrictions.

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
