# Peer review of "Convalescent Plasma Efficacy in Life-Threatening COVID-19 Patients Admitted to the ICU: A Retrospective Cohort Study"

_jcm, 2021, doi:10.3390/jcm10102113_

Round 1

Reviewer 1 Report

First of all, congratulations for this work on an unsolved subject. Your manuscript is very clear. The mian strength of this study is the relatively short time between the onset of symptoms and the administration of convalescent plasma. The statistics seem robust and the bibliography quite complete.

However, I have a few questions for the authors:

1- Why did some patients receive convalescent plasma and others not? Was it solely based on the availability of treatment, or at the discretion of the clinician in charge of the patient?

These clarifications seem necessary to me, especially since there is an important difference in treatment between the two groups concerning the use of non-invasive ventilation. Its more frequent use in the group receiving convalescent plasma could reflect a desire to avoid invasive ventilation. The question that arises behind this is: was the population in the convalescent plasma group more fragile (on criteria other than those analyzed), and is this why the physicians chose to treat them with convalescent plasma? Please complete the Method Section and the Discussion section with more details on that point. 

2- You do not provide information on the immune status of patients. I assume that they were all immunocompetent but it would be better to specify if they had solid cancers, hematological diseases or if they were under immunosuppressive treatment for dysimmune diseases or organ transplants. Please complete the Results section.

3- It may seem surprising that such a large proportion (almost 85%) of patients was treated with both convalescent plasma to help the immune response and with tocilizumab whose mechanism of action is immunomodulation. To my knowledge, this is the only study in this case. This point could have influenced the results and therefore deserves to be discussed. In this respect, it would have been interesting to specify the temporality of tocilizumab treatment in relation to the administration of convalescent plasma.

Author Response

Reviewer: First of all, congratulations for this work on an unsolved subject. Your manuscript is very clear. The main strength of this study is the relatively short time between the onset of symptoms and the administration of convalescent plasma. The statistics seem robust and the bibliography quite complete.

Response: We thank the reviewer for this supportive comment.

Reviewer: 1- Why did some patients receive convalescent plasma and others not? Was it solely based on the availability of treatment, or at the discretion of the clinician in charge of the patient?

These clarifications seem necessary to me, especially since there is an important difference in treatment between the two groups concerning the use of non-invasive ventilation. Its more frequent use in the group receiving convalescent plasma could reflect a desire to avoid invasive ventilation. The question that arises behind this is: was the population in the convalescent plasma group more fragile (on criteria other than those analyzed), and is this why the physicians chose to treat them with convalescent plasma? Please complete the Method Section and the Discussion section with more details on that point. 

Response:   We thank the reviewer for this important point. Convalescent plasma (CP) was given according to the hospital protocol for COVID-19 patients management during that period of time. According to our hospital protocol during the study period, all patients with COVID-19 pneumonia requiring low oxygen therapy, high flow oxygen therapy, non-invasive ventilation, or invasive mechanical ventilation were candidates to receive CP. Thus, basically, all our included patients were eligible to receive CP. However, the availability of CP lacked at that period of time, and not every eligible patient (based on the ABO type) was able to receive this treatment. Therefore, the decision of giving CP was not based on the severity or fragility of the patient but on its availability only. As shown in Table 1, SOFA and SAPS II scores were not significantly different between the two groups. Also, comorbidities distribution did not differ between the two groups. We agree with the reviewer that the rate of non-invasive ventilation was significantly higher in the CP group than in the other group. However, the severity of hypoxemia was the same in the two groups (PaO2/FiO2 ratio did not differ between the two groups, Table 1). We believe that the difference between the two groups regarding the rate of non-invasive ventilation is due to the observational design of the study. The decision of intubation was at the discretion of the clinician in charge of the patient and not based on the CP treatment.  Also, we adjusted for this difference in the multivariable analysis (table 2).

We have now added in the Method section, page 3:” The decision of CP administration was based on our hospital protocol for COVID-19 management. Patients with severe COVID-19 infection requiring low oxygen therapy, HFNO, NIV, or IMV were candidates for CP treatment. Exclusion criteria were patients under age of 18, pregnant women, and known IgA deficiency.” We also added in the Discussion section, page 10 (limitations of the study):” Also, many patients that were candidates for CP therapy did not receive the treatment because of lack of availability (based on ABO blood type matching between donors and patients).”

We hope that we have addressed the reviewer’s concerns appropriately.

Reviewer: 2- You do not provide information on the immune status of patients. I assume that they were all immunocompetent but it would be better to specify if they had solid cancers, hematological diseases or if they were under immunosuppressive treatment for dysimmune diseases or organ transplants. Please complete the Results section.

Response: We thank the reviewer for this point. Yes, the reviewer is right, all our patients were immunocompetent. None had any kind of cancer or was on immunosuppressive treatment. We have now added in the Results section, page 4:” All our included patients were immunocompetent, none had any cancer or was on immunosuppressive treatment.” We hope that we have addressed the reviewer’s concern appropriately.

Reviewer: 3- It may seem surprising that such a large proportion (almost 85%) of patients was treated with both convalescent plasma to help the immune response and with tocilizumab whose mechanism of action is immunomodulation. To my knowledge, this is the only study in this case. This point could have influenced the results and therefore deserves to be discussed. In this respect, it would have been interesting to specify the temporality of tocilizumab treatment in relation to the administration of convalescent plasma.

Response: We thank the reviewer for this important comment. Tocilizumab was largely available in our hospital and was part of our hospital protocol for COVID-19 treatment. The criteria for Tocilizumab treatment were severe COVID-19 pneumonia with bilateral involvement requiring oxygen therapy and associated with interleukine-6 ≥ 50 ng/L and two of the following: temperature > 38°C, C-reactive protein ≥ 50 mg/dL, or ferritin ≥ 500 µg/L. There were also many exclusion criteria such as elevated transaminases, thrombocytopenia, active bacterial infection, neutropenia, etc. Many patients received Tocilizumab before CP, as the median time from ICU admission to tocilizumab was 0 days [IQR:0-1 days]. We are aware of the recent publication (RECOVERY trial) and the association between tocilizumab and lower mortality rate. However, we do not believe that the high rate of Tocilizumab administration would have influenced our results. Indeed, the distribution of Tocilizumab treatment was equal between the two groups (84.4% in the CP group vs. 85.9% in the non-CP group, p = 1.00, Table 1). Also, we have adjusted for Tocilizumab treatment in the multivariable Cox regression analysis (Table 2), and it was used to determine ipw estimators (Table S3. We have now added in the Discussion section, page 10:” A large proportion of our patients (85.5%) received tocilizumab (anti-interleukin-6 receptor) according to our hospital protocol treatment for COVID-19 patients (pneumonia with bilateral involvement associated with elevated interleukin-6 and C-reactive protein, requiring oxygen therapy). Recently, in a randomized, controlled, and open-label trial (RECOVERY), tocilizumab was associated with improved survival and other clinical outcomes in severe COVID-19 patients [28]. However, we do not think that the high prescription rate of tocilizumab would have influenced our findings. Indeed, tocilizumab was equally distributed between the CP and non-CP groups (Table 1), and it was adjusted for in the multivariable Cox regression analysis (Table 2) and used to determine ipw estimators (Table S3).“ We hope that we have addressed the reviewer’s concern appropriately.

We thank the reviewer for his constructive comments that have improved the quality of the manuscript.

Reviewer 2 Report

Thank you for the opportunity to review the manuscript on “Convalescent plasma efficacy in life-threatening COVID-19 patients admitted to the ICU”. The study aims to determine whether CP is associated with better clinical outcomes among severe COVID-19 patients. The problem background was well addressed. Convalescent Plasma (CP) treatments for COVID-19 were known to be not effective. The present study did not add any new pieces of information for the reader.

These are the comments for the authors

  1. What are the criteria for a patient being selected whether to receive CP or not during their ICU admission?
  2. Did the patient serum antibody titres were measured before CP administration? If not, how authors differentiate the post CP titre raised was due to the patient's self-response or it raised was due to the CP?
  3. There was a significant difference in the time of symptoms to ICU stay between the CP and non-CP groups. Did the CP who received had a more serious disease, to begin with, than the non-CP group?
  4. RCT may be too ideal and not feasible in some settings, this study should be at least carried out as a case-controlled study instead. The study was mainly reporting the retrospective findings and it suffered from serious selective bias. Thus, the results and conclusion are over claimed and invalid.

Author Response

Reviewer: Thank you for the opportunity to review the manuscript on “Convalescent plasma efficacy in life-threatening COVID-19 patients admitted to the ICU”. The study aims to determine whether CP is associated with better clinical outcomes among severe COVID-19 patients. The problem background was well addressed.

Response: We thank the reviewer for this supportive comment.

Reviewer: Convalescent Plasma (CP) treatments for COVID-19 were known to be not effective. The present study did not add any new pieces of information for the reader.

Response: We thank the reviewer for this comment. We agree with the reviewer that there is enough evidence that CP therapy is not effective in moderate to severe COVID-19 disease, but not in critically ill patients requiring high flow oxygen therapy (HFO), non-invasive ventilation (NIV), or invasive mechanical ventilation (IMV). That is why we respectfully disagree with the reviewer that our study does not add new information for the reader. Indeed, we have already explained in the Introduction and Discussion sections that only a very few studies (2 or 3) focused specifically on these patients with life-threatening COVID-19 disease. Among these studies, information regarding the antibody titer levels in the infused CP was not missed, or the time from symptoms’ onset to CP administration was too long (median ≥ 10 days). The particularity of our study is that we included only critically ill COVID-19 patients requiring HFO, NIV, or IMV, with known antibody titer levels in the administered CP and shorter time from symptoms’ onset to CP infusion (median of 6 days). These criteria were not in any previously published study. That is why we strongly believe that our findings add new information to what is already know on this topic. Of note, there is an ongoing randomized multicenter trial (CONFIDENT trial) to answer the question about the efficacy of CP therapy in this specific population of critically ill COVID-19 patients. Thus, the answer to this question is not resolved yet. That is why we concluded, page 11: “Until the ongoing RCT results [26], we do not recommend using CP in critically ill COVID-19 patients.” We hope that we have addressed the reviewer’s concern appropriately.

Reviewer: 1. What are the criteria for a patient being selected whether to receive CP or not during their ICU admission?

Response: We thank the reviewer for this important point. Convalescent plasma (CP) was given according to the hospital protocol for COVID-19 patients management during that period of time. According to our hospital protocol during the study period, all patients with COVID-19 pneumonia requiring low oxygen therapy, high flow oxygen therapy, non-invasive ventilation, or invasive mechanical ventilation were candidates to receive CP. Exclusion criteria were patients under age of 18, pregnant women, and known IgA deficiency. Thus, basically, all our included patients were eligible to receive CP. However, the availability of CP lacked at that period of time, and not every eligible patient (based on the ABO type) was able to receive this treatment. We have now added in the Method section, page 3:” The decision of CP administration was based on our hospital protocol for COVID-19 management. Patients with severe COVID-19 infection requiring low oxygen therapy, HFNO, NIV, or IMV were candidates for CP treatment. Exclusion criteria were patients under age of 18, pregnant women, and known IgA deficiency.” We also added in the Discussion section, page 10 (limitations of the study):” Also, many patients that were candidates for CP therapy did not receive the treatment because of lack of availability (based on ABO blood type matching between donors and patients).” We hope that we have addressed the reviewer’s concern appropriately.

Reviewer: 2. Did the patient serum antibody titres were measured before CP administration? If not, how authors differentiate the post CP titre raised was due to the patient's self-response or it raised was due to the CP?

Response: There is some misunderstanding. We did not measure patients’ serum antibody titers. We measured the antibody titers in the infused CP. The measurement of antibody titers in the CPs was done after the CPs were administered to the patients. This was clearly explained in the manuscript in the “Antibody titer determination” section. These were already stated as part of the study limitations: “Second, CP antibody titer assessment was performed after the plasma was administered to the patients” and “Fourth, there was no measurement of antibody titer levels of neutralizing antibody titers against SARS-CoV-2 in the recipient population before receiving CP.” We hope that we have addressed the reviewer’s concern appropriately.

Reviewer: 3. There was a significant difference in the time of symptoms to ICU stay between the CP and non-CP groups. Did the CP who received had a more serious disease, to begin with, than the non-CP group?

Response: We thank the reviewer for this comment. The difference in the median time from symptoms’ onset to ICU admission between the CP and non-CP groups can be explained by the observational design of the study and not related to the severity disease. Indeed, as shown in Table 1, SAPS II and SOFA scores were significantly different between the two groups. Also, comorbidities, degree of hypoxemia (PaO2/FiO2 ratio), lactate levels, invasive mechanical rate, use of vasopressors, renal replacement therapy rate were all not significantly different between the two groups (Table 1). Therefore, patients who received CP treatment did not have a higher disease severity than those who did not receive CP.  We hope that we have addressed the reviewer’s concern appropriately.

Reviewer: 4. RCT may be too ideal and not feasible in some settings, this study should be at least carried out as a case-controlled study instead. The study was mainly reporting the retrospective findings and it suffered from serious selective bias. Thus, the results and conclusion are over claimed and invalid.

Response: We thank the reviewer for this comment. We agree with the reviewer that RCT is complicated to be performed in a pandemic setting. The reviewer advised us to perform a case-control study, and this what we exactly did. We identified patients who had clinical improvement (cases) and those who did not (controls) and determined if the exposure (CP therapy) was associated with the outcome in univariate analysis (Table S1) and multivariable analysis (Table 2).  We agree that the retrospective design is an important limitation of our study, and we already acknowledged that in the limitations paragraph in the Discussion section, page 10. However, we respectfully disagree with the reviewer that our results and conclusion are over claimed and invalid. Indeed, we did a multivariable regression analysis and adjusted for many potential confounders (Table 2) found in univariate analysis (Table S1). Also, we used the inverse probability weights (IPW) with robust standard error to balance the two groups (CP and non-CP) on the potential confounding variables and estimated the average treatment effect of the CP. All these methods were used to minimize the effects of selection bias. Both analyses resulted in the same finding that CP was not significantly associated with time to clinical improvement in critically ill COVID-19 patients. We hope that we have addressed the reviewer’s concern appropriately.

We thank the reviewer for his constructive comments that have improved the quality of the manuscript.

Reviewer 3 Report

This is an excellent report on the use of CP in critically ill patients.  While this use case has faded over the course of the pandemic, the experience described in this report is useful.  I am confused a bit by table 2 which seems to suggest some evidence of efficacy (P 0.14) and given the small N, the lack of alternatives and emerging ideas about dosing and titer this might be more positive than first glance. So more clarity on table 2 would be helpful.

This paper may also be of interest.

https://www.medrxiv.org/content/10.1101/2021.03.12.21253373v1

See also this meta-analysis, your ref 14 was based almost entirely on press reports of the RECOVERY data which have subsequently been updated and show a trend for efficacy in less sick patients.

https://www.medrxiv.org/content/10.1101/2021.04.01.21254679v1

https://www.mayoclinicproceedings.org/article/S0025-6196(21)00140-3/fulltext

Author Response

Reviewer: This is an excellent report on the use of CP in critically ill patients.  While this use case has faded over the course of the pandemic, the experience described in this report is useful. 

Response: We think the reviewer for this supportive comment.

Reviewer: I am confused a bit by table 2 which seems to suggest some evidence of efficacy (P 0.14) and given the small N, the lack of alternatives and emerging ideas about dosing and titer this might be more positive than first glance. So more clarity on table 2 would be helpful.

Response: We thank the reviewer for this comment. We understand the reviewer’s point. However, we do not think that our data suggest some evidence of efficacy. A P-value of 0.14 is far away from the statistical significance cutoff value of 0.05. Also, the 95% confidence interval is too broad (0.23-1.22) (Table 2). Furthermore, the average treatment effect of CP based on the inverse probability weights (IPW) with robust standard error was not statistically different between the CP and non-CP groups with a p-value of 0.93. Also, Figure 1 shows no separation between the two curves even though it was only a univariate analysis. Overall, from a statistical standpoint, our findings do not suggest some evidence of efficacy of CP treatment. We hope that we have addressed the reviewer’s concern appropriately.

Reviewer: This paper may also be of interest.   https://www.medrxiv.org/content/10.1101/2021.03.12.21253373v1

Response: We thank the reviewer for sharing this study. This RCT mainly included moderate to severe COVID-19 patients with very few requiring invasive mechanical ventilation (13%). The median time from symptoms onset to randomization was 9 days. The authors did not find a significant association between CP use and clinical improvement within 28 days, which was the primary outcome. The authors found a significant reduction in 28-day mortality associated with CP. However, since mortality was not the primary outcome, this finding should be taken with precaution. This RCT resembles other previously published RCTs. We do not think that adding this RCT would be relevant to our study as it did not focus on critically ill COVID-19 patients. Also, we are trying to reference peer-reviewed studies as much as we can. However, if the reviewer thinks that it is important to add this trial to our references, we would be happy to do so.

Reviewer: See also this meta-analysis, your ref 14 was based almost entirely on press reports of the RECOVERY data which have subsequently been updated and show a trend for efficacy in less sick patients. https://www.medrxiv.org/content/10.1101/2021.04.01.21254679v1. https://www.mayoclinicproceedings.org/article/S0025-6196(21)00140-3/fulltext

Response: We thank the reviewer for this comment. We agree that the meta-analysis referenced in our study (ref. 14) was based essentially on press reports of the RECOVERY trial that has been now published in a pre-print version (https://www.medrxiv.org/content/10.1101/2021.03.09.21252736v1.full.pdf) and confirmed the findings reported in the meta-analysis (ref.14). We agree with the reviewer that CP might be effective in less sick patients or seronegative patients. We have now added the reference shared by the reviewer (https://www.medrxiv.org/content/10.1101/2021.04.01.21254679v1) with the following statement in the Discussion section, page 10:” In this regard, it has been suggested that CP might provide at least a modest benefit in seronegative COVID-19 patients [30].” 

The main caveat of the meta-analysis (https://www.mayoclinicproceedings.org/article/S0025-6196(21)00140-3/fulltext) is that it did not include the RECOVERY trial (https://www.medrxiv.org/content/10.1101/2021.03.09.21252736v1.full.pdf) with 11,558 patients. The second issue is that it included RCT trials (10 RCTs, n= 1417) but also observational studies (20 studies, n= 11,477). Focusing only on the 10 RCTs, there was no significant association between CP and mortality. However, when all studies were analyzed together, CP was associated with a significant reduction in mortality. Again, the RECOVERY trial (n= 11,588 patients) was not included in this meta-analysis. That is why we do not think it is of interest to add this meta-analysis to our study. However, if the reviewer thinks it is relevant for our study, we would be happy to satisfy her/his request. 

We thank the reviewer for his constructive comments that have improved the quality of the manuscript.

Round 2

Reviewer 2 Report

Thank you again for the second time' review of the author's rebuttals.

First of all, the study design did not reflect that it was a case-control study.

The authors may have mistaken about retrospective cohort study with the case-control study. In fact, they presented their study in a cohort manner which may lead to a systematic error in the study design. The detailed matching of cases was not reported in the manuscript. Comparing both groups (CP vs Non-CP) in demographic data, even though looks comparable, did not guarantee it is a proper case-control study design. 

Be aware that the term ‘case-control study’ is frequently misinterpreted. All studies which contain ‘cases’ and ‘controls’ are not necessary all case-control studies. One may start with a group of people with a known exposure and a comparison group (‘control group’) without the exposure and follow them through time to see what outcomes result, but this does not constitute a case-control study.  Proper matching of each case and control with a minimal ratio of 1:2 is good but it is not as simple as that. The matching should include well-defined parameters for each case and control, which the authors missing the control definition in the methodology section. The authors claimed that the study is a case-control study but yet they didn’t provide detail matching criteria for each group which makes the study a merely retrospective cohort study and the conclusion may be over justified.

The over claimed conclusion was referred to this statement “Until the ongoing RCT results, we do not recommend using CP in critically ill COVID-19 patients” which is a strong recommendation. The sample size calculation was missing, which is crucial before the author makes a significant recommendation. The authors may use The Grading of Recommendations Assessment, Development and Evaluation (GRADE) approach for treatment recommendations which is widely available to gauge this recommendation.

Minor comments

  1. “Antibody titer against SARS-CoV-2 assessment in the CP was done after CP was administered to the patients. Seven donors were found to have CP with negative antibody titers, and the patients who received those products were excluded from the analysis.” These two statements were confusing when read together. The author shall provide information on the 12 (37.2%) CP titer which is below 1:160 that were given to the patients. This information is crucial to clear out these confusions.
  2. The authors assessed clinical improvement outcomes based on the WHO 6-point disease severity scale on ICU admission. There was more scale 5 patients than scale 4 in group NCP as compare with the CP group, which were inversely proportionate. This appeared that more ill patients in a Non-CP group as compare to the CP group. Author shall remove this p-value because not supposed to be calculated and it didn’t make sense, to begin with. The Chi-square analysis shall not be used to calculate the comparison.
  3. The authors shall provide the mean score of clinical improvement after the CP was given in the demographic data section.
  4. The authors shall explain in the methodology section the timing of the blood tests were taken and tested.
  5. I am well informed about the limitation that the patients' serological titers were not measured which indeed may affect the paper quality and significants.

I could not be more positive until the study was done with a proper case-control design because this involved important recommendations or rejection of new intervention for a critically ill patient. Or the authors may report the finding as a case series that makes the study scientifically sound.

Author Response

Reviewer: First of all, the study design did not reflect that it was a case-control study. The authors may have mistaken about retrospective cohort study with the case-control study. In fact, they presented their study in a cohort manner which may lead to a systematic error in the study design. The detailed matching of cases was not reported in the manuscript. Comparing both groups (CP vs Non-CP) in demographic data, even though looks comparable, did not guarantee it is a proper case-control study design. Be aware that the term ‘case-control study’ is frequently misinterpreted. All studies which contain ‘cases’ and ‘controls’ are not necessary all case-control studies. One may start with a group of people with a known exposure and a comparison group (‘control group’) without the exposure and follow them through time to see what outcomes result, but this does not constitute a case-control study. Proper matching of each case and control with a minimal ratio of 1:2 is good but it is not as simple as that. The matching should include well-defined parameters for each case and control, which the authors missing the control definition in the methodology section.

Response: We thank the reviewer for this comment. When we replied that our study was a case-control study, it was for 2 reasons: first, cases and controls were chosen from the same almost homogenous critically ill COVID-19 population admitted to ICU and required high-flow oxygen therapy, non-invasive ventilation, or invasive mechanical ventilation; second, as we already mentioned in the statistical analysis paragraph, we used the inverse probability weight method to balance/match the CP and non-CP groups on the pre-determined potential confounding factors (age, gender, body mass index, comorbidities, SOFA/SAPS II scores, IMV, renal replacement therapy (RRT), ECMO, use of vasopressors, steroids, and tocilizumab) and determined the average treatment effect of the CP in this matching population. We presented in Table S3 the weighted standardized differences of the confounding variables. We also stated that “There is no rule regarding how much imbalance is acceptable in a propensity score. Proposed maximum standardized differences for specific covariates range from 10 to 25% [19-21].” We agree with the reviewer that the design of our study is not typically a case-control design in the truest sense of the word as we did not pair individual “case” to individual “control” on age, gender, etc. However, we tried to balance the whole population on the different potential confounders using a matching technique (IPW). However, the reviewer is well aware that it is almost impossible to match on more than 4 or 5 factors individually in a case-control design. Also, case-control studies are not better or prone to fewer biases than retrospective cohort studies with multivariable analysis adjusting for potential confounding factors, especially when there are many possible confounding factors. Of note, a case-control design is different from a matched-control design as in the first one, each factor should be exactly paired between each case and each control (same age, same-sex, etc.), while in the latter design, a matching method is used to approximately balance the two populations on different variables. 

We agree with the reviewer that our study was a retrospective cohort than a typical case-control study. We did not mention case-control design in our manuscript. However, we did two sorts of analysis. The first one, a multivariable analysis (Cox regression) adjusting for the confounding factors to determine the association between CP and the primary outcome. The second one, IPW analysis to matches on the potential confounding factors (Table S3) and calculates the average treatment effects of the CP between the two groups. We hope that we have addressed the reviewer’s concern appropriately.

Reviewer: The authors claimed that the study is a case-control study but yet they didn’t provide detail matching criteria for each group which makes the study a merely retrospective cohort study and the conclusion may be over justified.

Response: We thank the reviewer for this comment. Case-control design is not better or less prone to biases than a retrospective cohort design (especially with multivariable analysis), as it seems suggested by the reviewer. Both designs are on the same level in the evidence base medicine pyramid. Also, as we mentioned in the previous response, it is quasi-impossible to individually pair cases and controls when there are many potential confounding factors. Case-control design is different from a matched group control design when a statistical technique of matching (propensity score analysis, IPW, etc.) is used to approximately match the two groups on pre-determined confounding factors. In the second part of our analysis, we used the IPW to balance the two groups on the potential confounding factors to determine the average treatment effect of CP. We hope that we have addressed the reviewer’s concern appropriately.

Reviewer: The over claimed conclusion was referred to this statement “Until the ongoing RCT results, we do not recommend using CP in critically ill COVID-19 patients” which is a strong recommendation. The sample size calculation was missing, which is crucial before the author makes a significant recommendation. The authors may use The Grading of Recommendations Assessment, Development and Evaluation (GRADE) approach for treatment recommendations which is widely available to gauge this recommendation.

Response: We thank the reviewer for this important point. We agree with the reviewer’s comment that the word “do not recommend” is strong. We have now deleted the following sentence: “Until the ongoing RCT results, we do not recommend using CP in critically ill COVID-19 patients.” We hope that we have addressed the reviewer’s concern appropriately.

Reviewer: Minor comments: 1. “Antibody titer against SARS-CoV-2 assessment in the CP was done after CP was administered to the patients. Seven donors were found to have CP with negative antibody titers, and the patients who received those products were excluded from the analysis.” These two statements were confusing when read together. The author shall provide information on the 12 (37.2%) CP titer which is below 1:160 that were given to the patients. This information is crucial to clear out these confusions.

Response: We are sorry, but we are confused by the reviewer’s request. Does the reviewer want information on the 12 donors that donated the CPs with titers below 1:160? What kind of information? How can this help clear out the confusion? And which confusion? Please clarify your comment, and we will be happy to respond accordingly.

Reviewer: 2. The authors assessed clinical improvement outcomes based on the WHO 6-point disease severity scale on ICU admission. There was more scale 5 patients than scale 4 in group NCP as compare with the CP group, which were inversely proportionate. This appeared that more ill patients in a Non-CP group as compare to the CP group. Author shall remove this p-value because not supposed to be calculated and it didn’t make sense, to begin with. The Chi-square analysis shall not be used to calculate the comparison.

Response: We thank the reviewer for this comment. We agree with the reviewer that it is confusing, and we have now removed the p-value according to the reviewer’s request.  We hope that we have addressed the reviewer’s concern appropriately.

Reviewer: 3. The authors shall provide the mean score of clinical improvement after the CP was given in the demographic data section.

Response: We are sorry, but the reviewer’s request is not clear. How long after the CP was given the reviewer wants us to provide the mean score of clinical improvement? For instance, on day 7, the median score of clinical improvement was 4.5 [IQR: 4-5] in the CP group and 5 [IQR: 3-5] in the non-CP group (p=0.60). On day 28, the median score of clinical improvement was 3.5 [IQR: 1-5] in the CP group and 2 [IQR: 1-5] in the non-CP group (p=0.16). Please let us know at what time the reviewer wants us to provide this information in the manuscript. We hope that we have addressed the reviewer’s concern appropriately.

Reviewer: 4. The authors shall explain in the methodology section the timing of the blood tests were taken and tested.

Response: We thank the reviewer for this comment. We have already mentioned in the methodology section that blood tests were collected on admission to ICU:” Laboratory values, including full blood count, coagulation parameters, and inflammatory markers (C-reactive protein, interleukin 6, and ferritin), were collected on admission to ICU.” We hope that we have addressed the reviewer’s concern appropriately.

Reviewer: 5. I am well informed about the limitation that the patients' serological titers were not measured which indeed may affect the paper quality and significants.

Response: We agree with the reviewer’s comment. We have acknowledged this in the limitations of the study. However, the serological test against SARS-CoV-2 was not available at that period of time in our hospital. This was not uncommon at the time of the treatment during the early phase of the pandemic. We hope that we have addressed the reviewer’s concern appropriately.

Reviewer: I could not be more positive until the study was done with a proper case-control design because this involved important recommendations or rejection of new intervention for a critically ill patient. Or the authors may report the finding as a case series that makes the study scientifically sound.

Response: We thank the reviewer for this comment. We hope that we were able to convince the reviewer that: 1) a proper case-control design is not feasible in the presence of many confounding factors; 2) a case-control design is not superior to a retrospective cohort design, mainly when multivariable analysis method is used to adjust for confounders; 3) A proper case-control design is different from a matched control group, which uses statistical techniques to approximately match two groups on potential confounders, which we did in part (IPW); 4) our study design (retrospective cohort), and statistical analysis (multivariable analysis adjusting for potential confounders and IPW analysis) are scientifically sound and have been used in many previously published studies.  

In a case-series design, there is no control group; all patients receive the treatment, which is not the case of our study. That is why it is not scientifically correct to report our findings as a case series.

Our study aimed not to provide recommendations on giving or not CP to critically ill COVID-19 patients. Neither case-control design nor prospective cohort design with or without matching will have the evidence to recommend or not to give CP to these patients. Only large RCTs or meta-analyses of RCTs would be able to provide solid evidence to recommend or not CP in these patients.  We aimed only to investigate the efficacy of CP in this specific population. We hope that we have addressed the reviewer’s concern appropriately.

We thank the reviewer for his constructive comments that have improved the quality of the manuscript.